# Negotiating illness through faith: Religious narratives of cancer etiology among patients and caregivers in Addis Ababa, Ethiopia

Kaleab Fikre ⓘ*, Abeje Berhanu Kassegne◉, Getnet Tadele◉

Department of Sociology, Addis Ababa University, Addis Ababa, Ethiopia

◉ These authors contributed equally to this work.
* kaleabfikre2@gmail.com

## Abstract

Cancer has become a major global public health threat, and individuals diagnosed with cancer and their family caregivers often seek to understand their illness experiences. Religious, spiritual, and sociocultural beliefs play a central role in shaping illness interpretations and care-seeking behaviors. However, in Ethiopia, limited attention has been paid to how these frameworks influence cancer experiences. This study explored religious narratives and interpretations of illness etiology among individuals diagnosed with cancer and their family caregivers and examined their influence on illness experiences and health-seeking practices. An interpretive phenomenological approach was employed using semi-structured, in-depth interviews with 41 participants, including individuals with cancer, family caregivers, and religious leaders. Data were transcribed, translated, and thematically analyzed to examine the processes of meaning-making. The findings showed that participants commonly conceptualized addiction as spiritually mediated rather than merely biologically determined. Illness was frequently interpreted as having a divine or supernatural origin, including punishment, a test of faith, God's will, or spiritual affliction. These interpretations were dynamic and shaped by religious teachings, sociocultural contexts and personal reflections. Religious frameworks influence emotional responses, coping strategies, and health-seeking behaviors, providing comfort and resilience while shaping treatment decisions. They functioned not only as explanatory frameworks for illness but also as practical resources that structured participants' responses to and negotiations with uncertainty, suffering, and responsibility. Religious meaning-making plays a central role in illness interpretations and overall experiences in cancer trajectories. Recognizing and engaging with these spiritual frameworks may enhance culturally responsive and patient-centered cancer care. Integrating spiritual sensitivity into clinical practice and collaborating with religious leaders may improve communication, trust, and psychosocial support for patients and their families.

**Data availability statement:** All relevant data are within the paper and its Supporting information files.

**Funding:** This study was supported by a PhD research grant awarded to the first author (Kaleab Fikre) from Addis Ababa University and Wolaita Sodo University. The funders had no role in study design, data collection and analysis, decision to publish, or preparation of the manuscript.

**Competing interests:** The authors have declared that no competing interests exist.

## Introduction

The rapid increase in cancer incidence is receiving global attention as a major public health threat and an increasing socioeconomic burden of the 21st century. In 2022, the World Health Organization (WHO) reported an estimated 20 million new cancer cases and 9.7 million cancer-related deaths [1,2]. In many low- and middle-income countries (LMICs), cancer is not only increasing rapidly but is also imposing a double burden on already fragile healthcare systems [3,4]. Although cancer incidence is lower in LMICs than in high-income countries (HICs), these regions experience a disproportionately high mortality rate compared to HICs [5]. This is due largely to late diagnosis, limited access to diagnostic and treatment services, and fragile health system capacity [1,4].

In the African Region, more than 900 000 new cancer cases and over 580 000 deaths were recorded in 2022, demonstrating the growing public health impact of the disease [5]. Ethiopia is also experiencing a rapid increase in cancer cases and mortality [5–7]. The WHO report showed that in 2022, an estimated 80,334 new cancer cases and 54,698 deaths occurred [8]. The national healthcare system is weak in effectively responding to this rapidly growing burden due to limited infrastructure, a shortage of trained personnel, and limited access to essential cancer services [9,10]. In addition, insufficient public awareness, sociocultural interpretations of illness, and unfavorable socioeconomic conditions influence care-seeking pathways [9,11,12].

From the onset of symptoms and throughout illness trajectories, patients and care-givers frequently seek to understand how cancer entered their lives. These questions often reflect existential concerns that cannot be fully answered or addressed within biomedical frameworks [13,14]. People attempt to make sense of the ultimate cause of the illness and the suffering it brings to the entire family. These reflections often manifest in deeply personal and existential questions such as, "Why me?", "Why us?", or "Why now?" [14]. In response, many draw on non-biomedical explanatory models, such as sociocultural and spiritual frameworks, to interpret illness, address these existential questions, and cope with uncertainty [14–16].

Previous studies have indicated that sociocultural beliefs and explanatory models play a central role in shaping how individuals and families interpret illness, seek care, and engage with treatment [15,17,18]. Religion, in particular, plays a crucial role in individuals' daily lives and in all cultures and societies [15,19,20]. Research suggests that religious beliefs and practices are an important source of values, meanings, and images for people seeking answers to existential questions concerning human vulnerability and destiny, especially in the context of serious illness [21,22]. In health crises such as cancer, religious frameworks may influence perceptions of disease causation, treatment options and coping strategies [20–22].

In low-resource settings, the strong influence of religious and spiritual practices is often linked to limited access to modern biomedical services, high medical costs, and geographical barriers [19,23]. Studies have shown that even individuals with access to modern healthcare may turn to religious practices as a primary or complementary source of support [20,24]. In Ethiopia, religion and cultural beliefs are crucial parts of social life, shaping values, daily practices, beliefs, and approaches to health and

illness [18–20]. Religious interpretations often inform understandings of suffering, guide treatment decisions, and guide coping strategies [18,19]. Importantly, engagement with religious practices does not necessarily reflect the rejection of biomedical services or as an option for limited healthcare services; rather, it expresses deeply rooted cultural identities and social relationships [23,25].

Despite the growing recognition of the importance of sociocultural and religious factors in health, biomedical systems have often given limited attention to patients' and caregivers' spiritual worldviews [20,26]. While some studies portray religious beliefs as barriers that delay biomedical care [27,28], others highlight their positive functions, including emotional support, resilience, and meaning-making [29]. Recognizing the significant influence of local belief systems in shaping illness interpretations and care-seeking behaviors, recent studies have proposed their incorporation into healthcare systems as a pathway toward more culturally responsive and patient-centered cancer care [29,30].

In Ethiopia, cancer research has examined epidemiology, focusing on prevalence and clinical outcomes, as well as limitations in healthcare infrastructure and service delivery [7,31–34]. These studies were usually based on biomedical perspectives and methodologies, paying less attention to the lived experiences of patients and family caregivers and the sociocultural and spiritual dimensions of illness interpretations, which can significantly influence illness experiences and cancer-care trajectories. In a context like Ethiopia, what patients and caregivers think and believe about the origin of illness, which is usually embedded in their sociocultural and religious or spiritual frameworks, is crucial as it significantly affects cancer trajectories. However, there is still limited understandings of how patients and caregivers make sense of cancer, negotiate diverse illness explanatory frameworks, including biomedical and religious frameworks, and orient their perspectives on cancer care practices.

This study explored religious narratives and interpretations of illness etiology among individuals diagnosed with cancer and their family caregivers in Ethiopia. Using an interpretive phenomenological approach, this study draws on in-depth interviews with patients, caregivers, and religious figures. It examines how participants understand the causes of cancer, construct meaning around illness, and incorporate spiritual beliefs into their healthcare decisions and coping strategies. By foregrounding lived experiences, this study contributes to a more context-sensitive understanding of cancer and provides evidence to inform culturally responsive healthcare practices and policies in Ethiopia and in similar contexts.

In this study, we employed multiple but complementary theoretical frameworks to assist in data analysis. First, we drew on Kleinman's explanatory model framework [35], which emphasizes how individuals and families interpret illness causation, symptoms, and treatment within cultural contexts. Second, it is guided by Berger and Luckmann's social constructionist perspective, which views the meanings of illness and suffering as socially produced through shared religious and cultural narratives. Third, in this study, we adopted Foster [36]'s personalistic etiology, recognizing that participants attributed cancer to spiritual or moral causes. Finally, we employed spiritual coping frameworks to examine how religion contributes as a coping resource for managing distress, uncertainty, and suffering, contributing to positive coping. Together, these perspectives provide an integrated conceptual framework for understanding how lived experiences of cancer are interpreted through religious meaning-making processes that shape treatment options, health care seeking decisions and coping strategies.

## Materials and methods

### Study design

This study adopted an interpretive phenomenological approach [37–39]. The interpretive phenomenological approach (IPA) focuses on understanding the lived experiences of study participants [39]. It is a philosophical approach founded on the ontological view that lived experience is an interpretive process rather than a descriptive one [37]. IPA involves a thorough examination of an individual's life world, exploring personal perceptions and experiences or account of an object or event, but does not attempt to produce an objective statement of the experiences, or object or event itself [38]. In the

context of this study, IPA was employed to explore narratives of religion-related illness etiology among individuals diagnosed with cancer and their family caregivers in Ethiopia.

## Researcher reflexivity and positionality

The corresponding author, who conducted the fieldwork and collected qualitative data, was a healthy male in academia with research training,. Hence, he occupied more stable social, educational, and institutional advantages than participants suffering from cancer and the caregiving burden, physically, emotionally, and economically. The researcher was in a better position of professional stability and physical well-being than the study participants, who had advanced illness, emotional uncertainty, and limited access to resources. These differences have had a significant impact on shaping interview dynamics, including the ways in which participants framed narratives, expressed suffering as patients and caregivers, or positioned the researcher as an authority.

Moreover, the researcher was aware of the influence of his sociocultural and religious background on shaping his interactions with the participants throughout the data collection period. This awareness allowed the researcher to be sensitive to the participants meaning of their suffering with cancer and attentive to the moral and spiritual meanings attached to cancer. At the same time, the researcher was aware that sharing a common culture or religious beliefs could lead to taken-for-granted assumptions or selective attention during data collection and interpretation.

To mitigate the influence of the researcher's positionality, reflexive practices were maintained throughout the research process by taking field notes, critically looking at emotional moments, and prioritizing the voices of study participants over the researchers' own interpretations. During the interviews, the researcher made efforts to exercise non-hierarchical interactions, allowing the study participants to express themselves and lead the narratives of their lived experiences. Similarly, during data analysis, the meaning participants attached to their experiences were given to researcher's own interpretations.

Thus, the researcher used reflexivity not as a limitation to be eliminated, but as an analytic resource to enhance transparency and rigor in interpreting the cancer experiences as they were narrated within unequal social, health, and institutional contexts.

## Settings

This study was conducted primarily at the Tikur Anbessa Specialized Hospital (TASH) in Addis Ababa, Ethiopia. TASH was chosen because it is the primary referral hospital in the country and receives thousands of cancer patients, providing comprehensive oncology services for patients seeking care from across Ethiopia and neighboring countries and housing the country's only radiation facility to provide radiotherapy. In this study, two non-governmental organizations (NGOs), namely the Tesfa Addis Parents Childhood Cancer Organization (TAPCCO) and the Mathiwos Wondu-Ye Ethiopia Cancer Society, were included to recruit participants who received cancer care at TASH with emotional, social, and financial support from the two NGOs. The two NGOs were included, taking into consideration their well-established relationships in supporting patients and their families in receiving cancer care at TASH. They primarily support patients and families who travel long distances seeking cancer care at TASH and provide various services during their hospital appointments, including accommodation, food, transportation services, and covering medical bills.

## Participants and recruitment

In this study, individuals diagnosed with cancer and their family caregivers were recruited using a purposive sampling technique based on their direct experience as patients undergoing biomedical treatment or as caregivers accompanying the patient in a hospital setting. Although the initial aim was to recruit a comparable number of

patients and caregivers, the study included fewer patients due to illness severity, as most presented at advanced stages, treatment schedules, and fatigue during treatment, particularly during and after chemotherapy. In contrast, caregivers were more accessible, interested, and actively involved in care decisions and spiritual practices. Although some caregivers were responsible for children with cancer, only adult caregivers were interviewed in this study. Information regarding the pediatric patients was provided by their caregivers. Given the study's focus on relational meaning-making within families, caregivers were retained as key participants. In addition, although the number of patient interviews was limited, an in-depth case-by-case analysis revealed strong convergence in core meaning-making patterns across participants. The final sample composition reflects both ethical considerations and fieldwork realities in the study setting.

Initial contact with the study participants was facilitated by an oncology nurse from TASH who had direct professional contact with patients and their caregivers during routine clinical visits. The corresponding author approached potential participants individually, informed about the purpose of the study, and invited to participate. Interested individuals were referred to the corresponding author for further information and consent procedures. Participants from the two NGOs were contacted through TASH's networks.

In this study, a total of 41 participants: 33 family caregivers, six individuals diagnosed with cancer aged between 19 and 64 years participated (see Table 1 for demographic characteristics of the study participants), and two religious fathers. Purposive sampling was used to recruit participants based on three inclusion criteria: 1) individuals diagnosed with cancer who were receiving cancer care at TASH during the data collection period; 2) family caregivers who accompanied individuals diagnosed with cancer during their hospital appointments; and 3) at least one month since cancer diagnosis. Individuals who met the criteria and expressed interest in the study were contacted in person and interviewed. In addition, the study included two religious fathers from the Ethiopian Orthodox Christian and Protestant Christian denominations.

Table 1. Demographic characteristics of study participants, N=41, 6 patients, 33 caregivers, 2 religious leaders.

| Characteristics | Patients (n=6) | Caregivers (n=33) | Religious leaders (n=2) | Total (N=41) |
|---|---|---|---|---|
| **Gender** | 2 | 19 | 2 | 23 |
| Men | 4 | 14 | – | 18 |
| Women | | | | |
| **Age (years)** | 2 | 22 | – | 24 |
| 19-45 | 2 | 10 | – | 12 |
| 46-60 | 2 | 1 | 2 | 5 |
| >60 | | | | |
| **Religious composition** | 1 | 19 | 1 | 21 |
| Ethiopian Orthodox | 3 | 7 | 1 | 11 |
| Protestant | 2 | 7 | – | 9 |
| Muslim | | | | |
| **Years since initial diagnosis** | 8 | – | – | 8 |
| >3months | 10 | – | – | 10 |
| 4-6months | 4 | – | – | 4 |
| 7months-1years | 17 | – | – | 17 |
| >1year | | | | |
| **Type of cancer** | 11 | – | – | – |
| Childhood cancer | 28 | – | – | – |
| Adult cancer | | | | |

## Data collection

Semi-structured, in-depth interviews will be employed to collect data from individuals diagnosed with cancer and their family caregivers from December 2024 to April 2025. Interview guiding questions were pre-prepared, allowing flexibility during the interview process to reorder questions, probe, and pose follow-up questions to collect rich, detailed data. The questions were developed for the purpose of this study, which is part of an ongoing research project informed by relevant literature and discussions with the two supervisors. The interview guide covered issues including participants' socio-demographic characteristics, interpretations of illness etiology, awareness of the disease, and care-seeking behavior of patients and their family caregivers. Questions related to religious narratives on illness experiences were not included as a pre-defined topic in the interview guide, as the original aim of the study was not specifically focused on religion. However, during data collection, participants frequently and spontaneously introduced religious interpretations when discussing their illness experiences. As the primary researcher, I (first author) recognized the importance of these accounts and explored them further through follow-up probes used only for clarification and deeper understanding. The focus on religious meaning-making therefore emerged inductively from the data rather than reflecting a change in the original study objective. The corresponding author conducted the interviews in Amharic, Ethiopia's official language. Two supervisors of this study, which is part of an ongoing Ph.D. research project, independently reviewed a subset of the transcripts to verify their accuracy.

The first three interviews with family caregivers took place in the corridors of the oncology center of TASH. The rest of the interviews were conducted in a quiet office space at TASH and the two NGOs. All the interviews were recorded with informed consent. Each interview lasted between 30 and 70 minutes and was audio-recorded with the consent of the participants. Observation data and informal dialogue with family caregivers were documented as field notes, allowing for triangulation with data from interviews. Participant recruitment continued until data saturation was reached, at which point additional interviews yielded limited new insights, and the dataset supported a coherent interpretive account [40]. Data saturation was assessed through an ongoing review of the interview transcripts and emerging codes. Saturation was reached after approximately 41 interviews with patients, caregivers, and religious figures, at which point further data collection was discontinued.

## Data analysis

Data collection and analysis were conducted simultaneously in this study. Following the procedures outlined by Smith et al. [38], we analyzed the data using interpretive phenomenological analysis. We transcribed audio-recorded data verbatim first in Amharic and then translated them into English and ensured that the information contained in the audio files accurately represented the transcripts. The analysis process involved reading the transcribed data repeatedly and importing them into ATLAS.ti 9 software for systematic coding. Initially, the first author conducted line-by-line coding, followed by the development of emergent themes, which were grouped into different categories, and the categories were subsequently clustered into specific themes. These themes were iteratively refined through constant comparisons across cases. The initial broad code was "perspectives or narratives on illness etiologies." This broader theme guided the initial data organization and was subsequently developed into subthemes. Reflexive memos were maintained throughout the analysis to document the analytical decisions and researcher assumptions.

## Ethical considerations

Throughout the research process and in all documentation, we ensured the confidentiality and anonymity of our participants by assigning random identifiers (e.g., participant 1, participant 2). In this study, we included only adult participants (aged ≥ 18 years), comprising adult patients, adult family caregivers, and religious leaders. No minors were directly interviewed or enrolled in the study. Prior to participation, we obtained informed consent outlining the purpose of the study,

procedures, and participants' rights during and after their participation. We assured our participants that their personal accounts would remain confidential and would not be disclosed to third parties. We also conducted the study in accordance with the ethical principles of the Declaration of Helsinki and received ethical approval from Addis Ababa University, Clinical Oncology Department (Approval No. 6/12/2024), and from the Institutional Review Board of the Ethiopian Society of Sociologists, Social Workers, and Anthropologists (IRB/ESSSWA, Meeting No. O26/12/2024).

## Trustworthiness of the study

We ensured the trustworthiness of the study through multiple, complementary strategies. Credibility was enhanced through prolonged engagement with participants and the triangulation of narratives across patients and caregivers, allowing for the identification of convergent and divergent perspectives. Dependability was supported by maintaining a clear audit trail, including detailed documentation of data collection procedures, coding decisions, and analytic processes. Confirmability was strengthened through ongoing reflexive journaling, which enabled critical reflection on the researchers' assumptions and potential influences on data interpretation. Transferability was addressed by providing rich, contextualized descriptions of the study setting, participants' sociocultural backgrounds, and the healthcare context, enabling readers to assess the relevance of the findings to other settings.

## Results

### Religious Narratives of Illness

Participants' accounts revealed that religious interpretations were central to how cancer was understood and experienced. Rather than being interpreted solely through a biomedical lens, cancer was frequently framed within spiritual narratives that shaped both meaning-making and care-seeking practices. These narratives included interconnected interpretations of cancer as a form of divine punishment for sins, a test of faith, a manifestation of divine or sovereign will, and the result of attacks by evil spirits or Satan. While analytically distinct, these interpretations often coexisted within individual accounts, reflecting the dynamic ways in which participants negotiated the meanings of illness.

Religious narratives of illness were not constructed solely through individual interpretation but were also shaped and reinforced through broader social and institutional influences, particularly religious leaders and community teachings. Participants' understandings of cancer were therefore embedded within shared moral and spiritual discourses circulating in religious settings. These narratives were reinforced during communal worship and pastoral counseling, contributing to their internalization and continued use in interpreting illness experiences. This broader religious context provides important grounding for understanding how participants developed, sustained, and negotiated meanings of illness.

Many participants attributed cancer to a divine or supernatural origin and drew on religious beliefs to make sense of suffering and uncertainty. For example, a participant caring for her 90-year-old grandmother reflected:

> We lived in Harar and moved to Addis Ababa after our father passed away. Since our father's death, there have been a series of deaths in our family. Therefore, I began to question why this is happening to us. Now, when something goes wrong or when there is no happiness at home, I often ask why, but not about illness. It is an effort to understand the meaning behind our suffering and losses (Participant 16, Caregiver).

This account illustrates how illness was interpreted within a broader history of loss and adversity, prompting ongoing existential reflection. Meaning-making extended beyond the illness itself, situating cancer within a cumulative narrative of suffering. In this context, religious beliefs provided a framework through which participants could render chaotic experiences intelligible and morally meaningful.

Across accounts, spiritual interpretations were frequently used to (re)frame illness experiences in relation to divine or supernatural forces, including God, Allah, Satan, or evil spirits. Religious texts and teachings, such as the Holy Bible and

the Qur'an, informed these interpretations, while personal and family experiences shaped how such meanings were constructed and sustained.

These narratives also had practical implications. Religious interpretations fostered coping by sustaining hope and enabling endurance during prolonged illness trajectories. At the same time, they influenced care-seeking behaviors, with practices such as holy water (*tsebel or zemzem*) used alongside—or at times in place of—biomedical treatment. This reflects a broader pattern of medical pluralism, in which religious and biomedical systems coexist and are actively negotiated. In this way, religion functioned not only as an explanatory framework but also as a practical resource shaping how patients and caregivers responded to diagnosis, treatment, and uncertainty.

**Cancer as a divine punishment.** A number of participants, including both individuals diagnosed with cancer and their family caregivers, interpreted cancer as a form of divine punishment linked to perceived sins or moral transgressions. These accounts reflected a process of moral and spiritual self-evaluation, in which participants located the origins of illness within their own actions or life histories. In this framing, cancer was understood as evidence of spiritual or moral failure and as a consequence of personal responsibility. More broadly, such interpretations facilitated ongoing reflection on past behaviors and reinforced practices of spiritual discipline.

This perspective is illustrated in the account of a participant caring for his son:

Cancer was brought by Allah. Is this something caused by humans? It could be that Allah is angry with me. It is Allah's anger. It could be to test me. It could be to bring me closer to Him. It could be to make His way clear. It could be His way of correcting my path, bringing me closer to Him (Participant 7, a Muslim father with caregiving responsibility for his 14-year-old son diagnosed with leukemia).

While Islamic teachings commonly emphasize illness as a test of faith, some Muslim participants extended this interpretation to include notions of punishment for sin or wrongdoing. This suggests that personal experience and existential reflection played a significant role in shaping how religious teachings were interpreted and applied in the context of cancer.

Similarly, a woman caring for her mother diagnosed with breast cancer reflected as follows:

What kind of anger is this? God, how are You punishing me? Why? Why are You punishing my mother like this, who spent her whole life as a devoted Christian and is so committed to her beliefs? What sins have we committed? It is very difficult. I don't know; I've been in conflict with God. Why is she being punished at this age? (Participant 13, a woman caring for her mother diagnosed with breast cancer, Orthodox Christian).

Among Christian participants, punishment narratives were often accompanied by spiritual struggle and existential questioning. In this account, the caregiver (the daughter) expressed difficulty reconciling her belief in divine justice with her mother's perceived moral and religious life history as a devoted Christian. This tension generated uncertainty and emotional distress, leading to a sense of conflict with God rather than acceptance of the illness. In this way, punishment narratives sometimes destabilized previously held religious understandings and intensified existential questioning.

A cancer patient reflected as follows:

I have no idea what brought cancer into my life. But I do have a feeling that I may have done something wrong in my life; related to living in accordance with fully obeying God. It could the wrath of the Creator (Participant 32, a woman diagnosed with colon cancer, Orthodox Christian).

Besides their religious interpretations of the origin of a cancer diagnosis, as a reflection of divine anger or punishment for sins or transgressions, participants' accounts also indicated diverse responses to their diagnosis.

 

Some participants, particularly Muslims, accepted their illness experiences, framing them as Allah's positive intention to draw them closer to Him and His way, as Participant 7 and others indicated. These kinds of responses reflect how participants, despite suffering, accepted it as a divine plan and a religious strategy to cope in submission to divine decree. Furthermore, it allowed participants to strengthen their faith by offering a meaning to their experiences, contributing for positive coping.

On the other hand, many Christian participants expressed existential questioning regarding the onset of cancer, asking why it had happened and how such suffering had entered their lives. This questioning was often accompanied by a spiritual struggle to reconcile their religious beliefs with the lived reality of illness. In attempting to make sense of this tension, participants engaged in ongoing meaning-making processes that reflected both faith-based reflection and emotional distress. These experiences were frequently associated with negative emotional responses, including confusion, anxiety, and uncertainty.

Participants' punishment narratives were also reinforced through messages conveyed by religious leaders, who framed illness within broader moral and spiritual discourses of sin, repentance, and divine discipline. These interpretations linked illness and suffering to perceived moral transgressions, reinforcing understandings of cancer as spiritually consequential. Participants often encountered such narratives within their religious communities during communal services and pastoral counseling, where illness was frequently interpreted through a moral lens. This contributed to the internalization of punishment-oriented explanations of illness within religious settings. Rather than functioning as direct theological instruction, these messages shaped the moral frameworks through which patients and caregivers interpreted their experiences.

One pastor's explanation aligns with the above interpretation:

God punishes his creation in different ways, from afflictions like boils to skin conditions. The Holy Bible is filled with stories that reflect God's anger and its manifestation through illness or suffering (Senior Pastor, Evangelical Christian).

Similarly, a religious father and theologian from Kidist Silase University reflected this form of interpretation as ደዌ ዘ ሀጥያት or illness due to sins or moral transgression and divine disciplining:

Scripture teaches us that disobedience and sins can lead to suffering or disease as a form of punishment. Illness serves as a call to repentance and moral correction (Orthodox Christian religious leader).

Importantly, religious based punishment narratives functioned as authoritative moral discourses that influence and shape patients and their caregivers' illness interpretations, reinforcing the perception that cancer was spiritually meaningful rather than merely biological conditions.

However, such interpretations also influenced patients' and caregivers' emotions and care-seeking behaviors. On one hand, it intensified feelings of self-blaming, responsibility, and vulnerability, contributing to negative coping. On the other hand, it led towards religious ritual practices such as repentance, prayer, and renewed devotion as coping resources for redemption, which usually contributes to positive coping.

**Cancer as a test of faith.** Some participants interpreted the origin of a cancer diagnosis as a divine trial or a test of faith rather than merely a biomedical condition. Both Christian and Muslim participants framed illness as a means through which God/ Allah tests believers' devotion, resilience, and spiritual commitment during periods of hardship. Muslim participants, particularly, tended to accept cancer diagnoses as a trial, interpreting it as a sign of the goodness or virtue that Allah perceived in them, despite the suffering that cancer brought to individuals and their family caregivers. In this construction, suffering became an opportunity to demonstrate faithfulness, commitment to spiritual purity, and moral endurance rather than evidence of divine punishment.

A Muslim participant illustrated:

> I do not see cancer as Allah's punishment; it is a test of faith that He (Allah) brought into our family, but not because of our sins or our wrongdoings. It is something Allah wants to see how we respond in our faith during suffering. We have no choices other than to accept it and stay faithful in our prayer for His forgiveness and help (Participant 3, father, caregiver for a 6 year old daughter, who was diagnosed with kidney cancer, Muslim).

Likewise, a Muslim woman diagnosed with breast cancer and undergoing chemotherapy at Tikur Anbessa Specialized Hospital reflected on how her interpretation evolved over time, seeing cancer as a form of punishment to a test of faith.

> In the beginning, I was so saddened when I heard a cancer diagnosis was confirmed thinking about the future of my children, and questioned Allah seeing it as a punishment, but later I come to a knowledge that it might happen because He (Allah) saw something good in me. Islam teaches that something might happen for a divine reason, if not, it might not have happened to me (Participant 31, woman diagnosed with breast cancer, Muslim).

This account illustrates how initial interpretations characterized by fear and uncertainty were gradually transformed into more affirmative religious meanings. As participant 31 and other stories show, meanings about cancer etiology are not static but change over time. In this regard, the emotional changes patients and caregivers experience during illness trajectories due to the meanings they gained through religious education played crucial roles. In other words, the role of religious teachings and personal reflections played a crucial role in facilitating this reinterpretation process.

Another participant similarly described this transition:

> For some times, I felt my diagnosis as God's punishment for my sins or wrongdoings. But, over the course of the illness, I came across to see it as a test of faith. I do not see my health situation is the result of God's punishment rather I accept it as how God wanted to see my faithfulness and how I endure despite suffering (Participant 37, women with cervical cancer, Orthodox Christian).

Participants' accounts illustrate that patients and caregivers suffering from cancer continuously negotiated and reframed the meaning they attached to the origin of their illness. The shift from punishment narratives into as a test of faith perspectives reflects an adaptive process through which individuals sought to restore coherence and hope in the face of uncertainty. Patients and caregivers moved between competing meanings and interpretations, from punishment to a test of faith perspectives, enabling them to settle emotionally and develop coping mechanisms accordingly. In the pathways, although many initially suffered from a complex range of emotional turmoil, such as anxiety, fear, confusion, reframing illness as a spiritual trial facilitated acceptance and motivated engagement with diverse forms of intervention. In this way, "a testing" perspective functioned as both symbolic resources for meaning-making and practical tools for psychological and spiritual resilience.

Participants' a test of faith narratives was also reinforced by religious leaders, who framed illness and suffering as divinely ordained tests. Based on scriptural references from the Holy Bible, religious figures framed illness as spiritually meaningful rather than merely biomedical condition. Within the Orthodox Christian tradition, this interpretation is often referred to as *däwé ze ʾeset* (ደዌ ዘ እሴት), denoting a form of illness that carries spiritual value and is understood as an opportunity for moral and religious growth.

A religious father and theologian from Kidist Silase University explained:

> There is suffering where value is placed on pain…loss and chronic illness…suffering like this offers individuals a chance to prove their faith and receive divine blessings.

Similarly, a senior Protestant pastor stated:

In the Holy Bible, people were tested with chronic health crises. Such sufferings were permitted by God to see how they remain faithful to God. We have seen God intervene in people's lives—even in cases of cancer. That is why, even through tears and questions, we must keep praying.

These religious fathers' narratives were grounded on the scriptural texts, which serves as a foundational framework for interpreting illness and suffering. In this narrative, illness experience is not presented as punishment for sins or wrong-doing but as a divinely ordained trial or a test of faith, in which the afflicted demonstrated faithfulness and ultimately rewarded. Through this illness narrative, religious leaders not only legitimized illness as spiritually meaningful and morally constructive, but also shaped patients' and caregivers' narratives to accept their suffering as a meaningful and a rewarding phenomenon.

A trial narrative encouraged participants to accept their situation and view spiritual practices such as patience, prayer, and perseverance as evidence of spiritual growth, which contributes to positive coping. In this way, participants interpreted illness as a way to demonstrate religious faithfulness, cultivating closer relationships with God. This framing often shaped the perception of illness as it possessed intrinsic spiritual value, promoting emotional wellness and maintaining optimism by linking present suffering to future divine reward.

Besides, a test of faith narrative also placed implicit expectations on patients and their caregivers to remain faithful and emotionally composed. While a trial narrative enabled emotional certainty and comfort, it usually discouraged emotions such as doubt, anger, or despair; as such emotions might be perceived as a spiritual weakness or a failure to endure the test.

Overall, a test of faith narrative functioned as institutionally grounded meaning-making resource that shaped patients and caregivers interpretations of illness causations. This perspective also learned through religious father spiritual sermons, counseling, and guidance. This transformed cancer as spiritually meaningful and rewarding trial, promoting endurance and devotion as morally rewarding. In this way, a trial narrative shaped patients and caregivers' interpretations of suffering with cancer, enabled to cope with uncertainty, and maintained religious identity throughout the illness trajectory.

**Cancer as a divine plan (sovereign power).** Another prominent illness narrative interpreted cancer as part of a divine sovereign plan. Participants' accounts indicated that people came to accept their suffering by aligning it with divine will, which was understood to govern all aspects of life. However, this acceptance was rarely immediate or straightforward. Rather, participants developed it gradually over the course of illness, which was influenced by emotional uncertainties, scriptural based teachings, and personal engagement with holy books. This gradual process allowed participants to move from existential questions and self-blaming toward accepting their health conditions and develop trust in divine purpose.

Religious teachings and scriptural stories from holy books played a crucial role as participants frequently drew illness narratives from holy texts to reframe their experiences and situate illness within broader religious beliefs and scriptural meanings, providing spiritual interpretative frameworks for suffering. Scripture-based stories enabled participants to reframe their illness experiences as part of a meaningful divine design.

A narrative from a woman caring for her son with a brain tumor and undergoing biomedical treatment at Tikur Anbessa Specialized Hospital illustrates this process:

During the diagnosis, I frequently questioned what sin I had committed that this anger and condemnation came to me. But later, I recalled a story of a man who was born blind since childhood from the Bible. In this story, people gathered and asked Christ whether the man's blindness was because of his mother's or father's sins. But Jesus Christ answered that neither of his parents had sinned; it was so that the glory of God could be revealed. That gives peace of mind. We are all sinners while on earth; if a man says, "I am righteous," he is a liar. It provided me with a shift from questioning to aligning our current problem with God's good intention, which might be for the glory of His name. So, what we have been doing to stay strong in our faith and maintain our prayer begging for his mercy and miraculous interventions (Participant 24, Protestant Christian).

This narrative demonstrates how participants actively moved more from one form of narrative to the other aligning their experiences with scriptural stories. Religious interpretation allowed participants to make a shift from feeling guilty about sins or moral self-blaming to accepting them as part of divine purpose. Such reframing allowed participants to see themselves from being subjects of divine anger to parts of divine's sovereign plan. The alignment of personal or family suffering with scriptural stories enabled participants to reconstruct their perspectives of illness experiences as spiritually meaningful. Such narrative therefore helped participants emotionally to settle and to move from guilt and moral self-evaluation toward spiritual confidence. Consequently, despite painful experiences, scriptural stories enabled sufferers to find comfort and develop hope and optimism, upholding that they might carry a divine sovereign purpose. Divine's sovereign plan narrative of illness symbolically functioned to safeguard the social dignity and emotional wellness of participants, countering stigma and self-blame.

Besides its symbolic functions giving meanings of illness experiences, illness as a divine plan narrative played crucial roles strengthening participants' engagement with religious resources. This framing facilitated participant to access religious coping practices and access to social support, such as prayer, as a coping resource, reinterpreting a cancer diagnosis as meaningful rather than arbitrary. Prayer, religious counseling, and faith-based scriptural supports functioned as a practical resources through which participants maintained their emotional certainty, spiritual wellness and commitment. In this way, narratives of illness as divine's sovereign purpose functioned as powerful meaning-making spiritual resource that provided participants existential comfort for their existential concerns and reinforce culturally embedded frameworks for interpreting suffering.

**Cancer as a spiritual attack of evil spirit, Satan, or Evil Eye, *Buda*.** In contrast to divinely ordinated condition, quite a few of our participants interpreted the origin of cancer as a form of spiritual attack. Within this narrative, a cancer diagnosis was attributed to an affliction caused by a spiritual attack by evil spirits, Satan, or Buda, the Evil Eye evil spirits. Participants who adopted this narrative emphasized that God, who understood as healer, protective, and compassionate, would not intentionally inflict suffering on human beings, as it contradicts His nature. Consequently, participants' externalized disease as the work of malevolent spiritual forces rather than a manifestation of divine will.

A participant expressed this view as follows:

> It is Satan who causes illness. God does not want you to suffer from disease. Instead of disease, I believe that God provides comfort and healing. We pray daily and cry, seeking His mercy and healing. God does not inflict suffering. However, Satan is the one who causes disease and suffering to family (Participant 35, mother diagnosed with colon cancer, Protestant Christian).

This account showed that interpretation of illness not as divinely caused but as spiritually adversarial. This narrative functioned to maintain an affirmative image of God while attributing blame toward supernatural adversaries. On one hand, this narrative reinforced participants' religious trust and moral confidence despite their suffering with cancer. On the other hand, this spiritual attack framework reinforced spiritual practices such as prayer, fasting, and collective worship as coping strategies for protection and healing.

Beyond explicitly religious explanations, some participants attributed illness etiology to socio-culturally rooted belief, particularly the concept of *Evil Eye, or Buda*. Within this framework, a cancer diagnosis was understood as resulting from harmful mystical forces transmitted through envy, jealousy, or malevolent gazes. Participants believed that certain individuals possessed the power and feared for their ability to cause misfortune, physical harm, or serious illness simply by looking at others, particularly among vulnerable groups such as children.

A father caring for his son with leukemia explained:

> Some people in our community said it might be caused by the Evil Eye or demonic influence. When evil forces remain on a person for too long, it is believed to change into cancer. We were told that if a person is attacked by evil spirit and

not treated in time, it can eventually mark as a physical illness such as cancer (Participant 8, father caring for son with leukemia, Muslim).

This understanding indicated how illness was interpreted as socio-culturally mediated, linking physical illness to interpersonal social relationships and community dynamics. Rather than being viewed as purely a divine act or random, or a biological condition, illness was interpreted by embedding it within moral and social relationships in which envy and spiritual vulnerability played crucial roles.

An attribution of a cancer diagnosis to a spiritual attack framework thus served multiple functions. First, it allowed participants to be protected from self-blame, shifting responsibility away from personal wrongdoing perspective, externalizing disease causations to malevolent gazes. Second, it promoted spiritual practices, such as prayer, bathing in or drinking holy water, *tsebel or zemzem,* as spiritual protection and coping strategies, sometimes as an alternative, or sometimes as a complement to biomedical treatments. Third, the spiritual attack perspective reinforces social cautiousness, making people to become more attentive to perceived threats within their social contexts.

Participants' narratives also demonstrate that spiritual attachment interpretations were shaped by broader societal meaning constructions that link illness to spiritual exposure, the evil eye, buda, and moral vulnerability. Such societal belief systems significantly influence how families suffering from cancer assess risk, seek support, and evaluate treatment options. This narrative often influences care-seeking behavior, usually leaning towards the use of spiritually based care. However, this narrative also negative affects social cohesion, causing fear, suspicion, and dependency on spiritually-based care.

Overall, interpreting illness by attributing it to spiritual attacks or evil spirits, Satan, or the Evil Eye, Buddha, served as a socio-culturally embedded meaning-making strategy that maintains participants' religious faith, seeks spiritual coping practices, and situates illness or suffering within moral and social relationships. This narrative usually provided participants with emotional comfort from self-blame and maintains spiritual hope. However, it also strongly influenced care-seeking behaviors, reinforcing people's non-biomedical understanding of disease causation and encouraging them to seek non-biomedical interventions accordingly.

## Discussions

This study unveiled how individuals diagnosed with cancer and their family caregivers interpreted illness etiology in Addis Ababa, Ethiopia, and indicated diverse religious interpretations of illness etiology. Participants conceptualized cancer as spiritually mediated, rather than merely biologically determined. They indicated a dynamic interplay between religious worldviews, illness, and interpretation across cancer trajectories. The spiritual framing of illness or cancer offered strategies in meaning-making pathways of participants' experiences, enabling them to negotiate uncertain experiences or existential concerns, assume moral responsibility, and hope in the context of cancer. The study revealed multiple religious-oriented illness narratives rather than a single coherent illness interpretation. In addition, illness narratives were not fixed; rather, they evolved throughout illness trajectories owing to interactions between personal experiences and interpretations, sociocultural contexts, and exposure to diverse religious meanings. In this respect, the findings extend previous studies [11,19,20,23,41,42] by illustrating the centrality of religion in illness interpretation and the fluid and contested nature of these explanatory models. Similarly, studies have demonstrated the influence of sociocultural contexts in shaping the interpretations of illness etiology [38].

In this study, religious interpretations such as "punishment," "test of faith," "divine plan," and "attack of evil spirits" are understood within established explanatory frameworks of illness meaning-making. It draws on from Kleinman's [35] explanatory model framework, which views illness interpretations as culturally shaped accounts through which individuals explain illness causation and suffering. It also employed a social constructionist perspective, which reflects that illness narratives are shared religious understandings produced and sustained within communities. These interpretations also

correspond to Foster's [36] *personalistic etiologies*, which attribute illness to spiritual, moral, or divine agents rather than to solely biological processes. Furthermore, within spiritual coping frameworks, religious interpretations that frame illness as a test or divine purpose are often associated with positive coping, whereas interpretations emphasizing punishment may reflect spiritual distress. These conceptual distinctions guided the analysis of the participants' narratives in this study.

In line with previous studies in low-resource settings [15,19,30], this study indicates the centrality of religion in shaping health-related meaning making. This also aligns with VanderWeele's [43] argument that religion functions as a powerful social determinant of health, shaping both meaning-making processes and health-seeking behaviors. According to this study's participants, understandings illness was based on broader religious interpretations that guided how illness was understood and managed. Religiously grounded interpretive frameworks offered moral, emotional, and spiritual structures through which patients and caregivers made sense of cancer and navigated uncertainty throughout illness trajectories.

The findings revealed that spiritual interpretations were the most dominant frameworks and unveiled participants' diverse religious explanations for illness as having a spiritual origin, either as divine punishment for sin, a test of faith, a manifestation of God's sovereign will, or an affliction brought about by evil spirits or the devil. This interpretation aligns with the studies by Vecchiato [44] and Østebø et al. [18], which emphasized the role of supernatural agents in the etiology of illness in Ethiopia. Similar perceptions have been widely documented in low-and middle-income countries (LMICs), particularly in Africa, where illness is often attributed to supernatural beings [15,19,23,45]. Liddell et al. [45] similarly observed that relationships with supernatural beings are often embedded in everyday life and are central to family wellbeing. These interpretations align with Foster's [36] concept of *personalistic etiology*, in which illness is understood as resulting from the intentional actions of the divine (e.g., God), spiritual (e.g., evil spirits), or human (e.g., enemies).

The findings further demonstrated another form of *personalistic* etiology, which links illness to socio-culturally rooted beliefs, particularly the concept of the *Evil Eye or Buda*. This view attributes physical illness to interpersonal relationships and community dynamics. Rather than being viewed as a divine act, random, or a biological condition, illness was interpreted by embedding it within moral and social relationships in which envy and spiritual vulnerability played crucial roles. In the Ethiopian context, Boylston reflected that *Buda* attacks might be intentional or inadvertent [46]. While *buda* are often described as animal spirits, their actions can be traced back to humans, some of whom are believed to be permanently associated with them. Research conducted in Pakistan has indicated parallel belief systems, where envy, particularly toward children, is thought to manifest as the evil eye and cause harm [47]. This perspective thus intersects illness etiology not only to an attack of spiritual agents, but also to social relationships, particularly feelings of envy and resentment within the community.

The study also highlights the social authority of religion in shaping knowledge, beliefs, meanings, and explanations of illness. In this regard, religious teachings emerged as major sources of interpretive frameworks, serving not only theological functions but also collective meaning-making roles that enable patients and caregivers confront existential threats such as cancer [15,19,23,48]. Although access to biomedical services is increasing, spiritually based illness interpretations remain highly influential in Ethiopia [48], indicating how religious epistemologies persist alongside modern medical systems.

The study also demonstrated that religious interpretations varied both between and within Christian and Islamic traditions. Although personal reflections shaped illness interpretations across both faiths, Islamic traditions mostly framed illness as a test of faith rather than as punishment. This finding is line with previous research showing that within Islam, illness is often regarded as a sign of divine love and spiritual refainment [15,42,49,50]. These variations highlight the interaction between formal religious doctrine and individual meaning-making processes.

The findings further demonstrate that illness was understood not only as a physical condition but also as a spiritual experience. In this regard, Kraus and Desmond [51] study shows that individuals facing serious illness actively engage in "lived religion," whereby spiritual meaning is constructed through everyday practices, personal reflection, and relational experiences rather than through formal doctrine alone. This perspective highlights how illness becomes an embodied and

existential condition through which individuals reinterpret their relationship with the divine, reconstruct identity, and navigate uncertainty. From an intersectional framework, religious identity interacted with other social, cultural, and familial contexts to shape illness interpretations [52]. In line with earlier studies [15,53], participants commonly conceptualized cancer as having spiritual origins, reinforcing the central role of faith in illness narratives. From a social constructivist perspective, religious beliefs, teachings, and leaders play a crucial role in shaping how illness is understood and experienced [54]. These actors contribute to the production and circulation of dominant narratives that influence patients' and caregivers' perceptions, decisions, and coping strategies.

Religious interpretations shaped not only causal explanations but also emotional and behavioral responses to illness. Participants frequently reported raising existential questions such as "Why me?" and "Why us?" as they sought meaning in suffering and divine justice. Similar patterns have been documented among individuals with chronic illness who use religious frameworks to negotiate uncertainty and distress [21]. These existential struggles were more commonly expressed among Christians, whereas Muslim participants more often framed illness as an accepted test of faith [54].

The findings further demonstrated that symptoms onset often led patients and caregivers into existential questions, usually resulting in emotional problems, such as confusion, anxiety, and frustration. In response, participants frequently engaged in spiritual practices, such as prayer, visiting sacred places to bathe in or drink holy water, consulting with religious leaders, and participating in communal rituals. Previous studies similarly indicated that religious participation serves as a key coping strategy during serious illness such as cancer [11,34,36].

On the other hand, religious interpretations of illness etiology, particularly a test of faith narrative, helped patients and caregivers to accept their suffering as divine's goodwill and emotional resilience. This form of experience was particularly evident among Muslim participants, for whom illness was commonly understood as a test of faith, which was an expression of divine will. Such illness interpretations usually contributed for positive coping through fostering patience throughout illness pathways and spiritual commitment, enabling individuals to tolerate uncertainty and suffering with a sense of divine purpose [49,50]. As noted by Ashy [49], religious faith provides both patients and caregivers with psychological strength and social support during periods of hardship. This interpretation frequently contributed to positive coping.

Nevertheless, religious grounded illness interpretations and meanings are not fixed. Instead, they change during illness pathways through personal reflection, social relationships, and exposure to multiple knowledge systems. This finding is also aligns with Kraus's [55] study that showed religious and spiritual beliefs are dynamic, often shifting over time as individuals reinterpret suffering, renegotiate their relationship with the divine, and integrate new forms of knowledge. This perspective resonates with the present findings, where although religious meanings remained central, patients and caregivers frequently combined them with biomedical explanations, reflecting a fluid and negotiated process of meaning-making. While Kraus's work is situated in a Western context, the present study similarly demonstrates that in Ethiopia, religious meaning-making is dynamic and relational, shaped through ongoing interactions with family, community, and religious institutions, rather than being a static set of beliefs. This finding aligns with recent studies documenting the gradual integration of biomedical explanations into traditionally spiritual illness narratives in Ethiopia [35,51,55]. For example, communities increasingly acknowledge factors such as heredity and environmental exposure while maintaining spiritual interpretations.

## Implications of the study

This study has some implications for oncology practices, public health policy frames, and future research in Ethiopia and similar contexts. First, the study illuminated how spiritual frameworks are central in illness interpretations of cancer etiology, making it crucial to integrate religious support in cancer care pathways. Therefore, it is important to work with religious leaders, and use their religious social settings and networks to enhance public health knowledge regarding cancer and provide spiritual counseling. Besides, it is important to have a curriculum on health and religion in training healthcare professionals, as it helps them to understand the co-existence of multiple perspectives on illness etiology and enhance

the provision of holistic and patient-centered care. This can enhance trust between patients, caregivers and providers in the cancer care pathways.

Second, the study highlighted how spiritual interpretation of illness etiology is dominant among patients' and caregivers' interpretations of illness. This might require acknowledging the contextual influence in cancer prevention efforts and early detection programs. Therefore, it is important to recognize local belief systems and design public health interventions accordingly. Moreover, improving the active community participation and using religious social networks enable to increase cancer awareness, reduce delayed diagnosis, and promote timely medical care.

Third, this study demonstrates the value of an interpretive phenomenological approach in eliciting deeply held narratives that shape illness interpretation and care-seeking behaviors. By foregrounding patients' and caregivers' lived experiences, the study provides a nuanced understanding of how meaning is constructed around cancer within a specific sociocultural context.

Building on these insights, future research could extend this work in several directions. First, longitudinal studies exploring illness narratives across different stages of the cancer trajectory would provide deeper insight into how meanings evolve over time and in response to changing illness experiences. Such work could inform the development of more culturally responsive and context-sensitive healthcare services. Second, further research is needed to examine the influence of religious narratives on care-seeking behaviors within broader sociocultural and economic contexts. A more detailed understanding of these dynamics would support the integration of patient and caregiver perspectives into the design of equitable and effective cancer care systems.

Finally, comparative research across diverse sociocultural and healthcare settings would help assess the transferability of these findings. Studies conducted in contexts with similar characteristics,such as strong religious influences and pluralistic healthcare systems as well as in more secular or highly biomedicalized environments, could clarify which aspects of illness meaning-making are context-specific and which may extend across settings. Such work would contribute to a more comprehensive understanding of how explanatory models of illness are negotiated in different contexts.

## Limitations of the study

This study has limitations. The findings of this study are grounded in the specific sociocultural and religious context in which participants live, where religious traditions play a central role in shaping understandings of illness, care-seeking practices, and moral interpretations of suffering. While this contextual depth is strength, it may limit the transferability of the findings to settings with different sociocultural environments, religious orientations, or healthcare structures. In particular, the explanatory models identified here may be less applicable in contexts where secular biomedical frameworks predominate or where patterns of healthcare access, literacy, and institutional trust differ substantially. As such, the illness narratives should be understood as contextually situated rather than broadly generalizable.

Second, this study is based on participants' self-reported accounts. This is sometimes subject to recall bias and social desirability bias. It illustrates how certain illness etiologies, particularly religious ones, are dominant in participants' narratives. This is due to the strong influence of spiritual illness narratives in shaping meanings attached to illness experiences. Similarly, participants might not pay attention to biomedical perspectives if they perceive them as secondary to religious care.

Third, this study was conducted with cross-sectional research design, which captured participants' perspectives at a certain point in time. This limits to understand how explanatory models may change during illness trajectory. Therefore, although it has been difficult to follow patients for longer periods, as cancer is a deadly disease, a longitudinal research could offer a better understanding on how illness etiologies shift over time or during illness trajectories. Besides, exploring the perspectives of religious father could offer further understanding on how religious teachings shape illness narratives.

## Conclusion

This study unveiled the crucial role religion played in participants' interpretations of illness and overall experiences in cancer trajectories. Religion offered spiritual frameworks in the meaning-making of illness etiology. The study demonstrated that interpretations of illness etiology were not only seen as a biological condition but also as spiritually interceded. Furthermore, illness interpretations usually shift during illness trajectories, from the perspective of divine punishment to a trial or divine purpose narratives. In this regard, the role of religious teachings, sociocultural contexts of patients and caregivers, and personal reflection were important shaping participants' interpretations of illness etiology.

Besides, the study indicated how interpretations of illness etiology shaped participants' emotions, strategies they employed to cope with it, and care-seeking behaviors. On one hand, these religious and spiritual worldviews impacted some to experience emotional turmoil, such as anxiety, confusion, and stress, raising existential questions and struggling to accept their situations, contributing to negative coping. On the other hand, these interpretations shaped others to accept it as meant to the goodness of their fate, fostering patience and spiritual commitment and enabling individuals to tolerate uncertainty and suffering with a sense of divine purpose, contributing to positive coping.

## Supporting information

**S1 File. Interview-transcripts.**
(DOCX)

## Acknowledgments

The authors would like to thank all study participants for their time, willingness, and openness in sharing their experiences, which were essential for this study. We are also deeply grateful to the religious leaders, healthcare professionals, and individuals who facilitated access and provided guidance during the research process. We acknowledge the support of Tikur Anbessa Specialized Hospital for enabling and facilitating the study. Finally, we extend our appreciation to supervisors and colleagues for their constructive guidance throughout the development of this work. We are also grateful to the reviewers for their insightful comments, which significantly improved the quality of the manuscript.

## Author contributions

**Conceptualization:** Kaleab Fikre, Abeje Berhanu Kassegne, Getnet Tadele.

**Data curation:** Kaleab Fikre.

**Formal analysis:** Kaleab Fikre, Abeje Berhanu Kassegne, Getnet Tadele.

**Investigation:** Kaleab Fikre.

**Methodology:** Kaleab Fikre, Abeje Berhanu Kassegne, Getnet Tadele.

**Resources:** Kaleab Fikre.

**Supervision:** Abeje Berhanu Kassegne, Getnet Tadele.

**Writing – original draft:** Kaleab Fikre, Getnet Tadele.

**Writing – review & editing:** Kaleab Fikre, Abeje Berhanu Kassegne, Getnet Tadele.

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
