## [Decision Letter · Decision Letter 0]

9 Feb 2026

PONE-D-25-58785Religion, Meaning, and Cancer: Illness narratives and interpretation among patients and family caregivers in EthiopiaPLOS One

Dear Dr. Fikre,

Thank you for submitting your manuscript to PLOS ONE. After careful consideration, we feel that it has merit but does not fully meet PLOS ONE’s publication criteria as it currently stands. Therefore, we invite you to submit a revised version of the manuscript that addresses the points raised during the review process. ================================

The manuscript needs to undergo through several revisions. While both reviewers were receptive to the article’s proposal, it requires several changes to be in conditions of acceptance. The manuscript has several typos and writing mistakes, that need to be addressed – since this is the first round, I will not request professional revision immediately, but this is something to consider if there are new rounds. Both reviewers pointed out several issues with the quantitative analysis, that you need to address as well. The use of citations of the Bible and the Qu’ran also needs to be rethought, because the manuscript is veering into theology and if your study has more theological concerns, I believe it could be submitted to another journal. A similar issue is with the religious leaders’ quotes, they are approaching these questions theologically (not to mention as authorities, they do not need to be anonymized), you must select their quotes that fit the proposal of the study better or let it for another study. Since there are several issues, the next round will determine if the manuscript might be suitable for publication or not.================================

We look forward to receiving your revised manuscript.

Kind regards,

Rafael Galvão de Almeida, PhD.

Academic Editor

PLOS One

Journal Requirements:

4. In the online submission form, you indicated that the datasets generated and/or analyzed during the current study are not publicly available because the study forms part of an ongoing PhD dissertation, but are available from the corresponding author on reasonable request.

5. Please update your submission to use the PLOS LaTeX template. The template and more information on our requirements for LaTeX submissions can be found at http://journals.plos.org/plosone/s/latex.

6. We note you have included a table to which you do not refer in the text of your manuscript. Please ensure that you refer to Table 2 in your text; if accepted, production will need this reference to link the reader to the Table.

Reviewers' comments:

Reviewer's Responses to Questions

**Comments to the Author**

1. Is the manuscript technically sound, and do the data support the conclusions?

Reviewer #1: Yes

Reviewer #2: Partly

2. Has the statistical analysis been performed appropriately and rigorously? 

Reviewer #1: Yes

Reviewer #2: N/A

3. Have the authors made all data underlying the findings in their manuscript fully available?

Reviewer #1: Yes

Reviewer #2: Yes

4. Is the manuscript presented in an intelligible fashion and written in standard English?

Reviewer #1: Yes

Reviewer #2: Yes

5. Review Comments to the Author

Reviewer #1: The manuscript addresses an underexplored dimension of cancer care in Ethiopia—religious interpretations and meaning-making among patients and caregivers. The topic is important and culturally relevant. However, the paper requires major revisions before it can meet the scientific and methodological. Key concerns relate to conceptual clarity, methodological rigor, analytic depth, and internal consistency.

- Q1: Conceptual Framework: Not Well Defined

Although the study uses phenomenology, there is no clear theoretical or analytic framework guiding how “religious meaning-making” is conceptualized.

Q2. Why this is a problem: The manuscript repeatedly uses terms such as punishment, test of faith, sovereign plan, supernatural, etc., but does not explain how these constructs are defined or distinguished in existing scholarship.

Q3.The analysis risks becoming descriptive rather than theoretically grounded.

Required Revision

Provide a conceptual model linking phenomenology, meaning-making, and religious explanatory models.

Clarify whether the study draws from: Kleinman’s explanatory models, Berger & Luckmann’s social construction, Personalistic/naturalistic etiologies, Spiritual coping frameworks

Q4. Methodological transparency is insufficient

Several methodological elements require clarification or strengthening.

1. Sampling strategy is not sufficiently justified.

Why were only 6 cancer patients included but 33 caregivers?

Phenomenology typically emphasizes deep exploration with fewer participants, not imbalanced large samples.

2. No explanation of saturation.

How was thematic saturation reached?

3. Translation procedure insufficiently described.

Back-translation?

Were multiple coders involved?

4. Data analysis lacks rigor.

No clearly mention of software, coding reliability, reflexivity, or audit trails.

Q 5: Over-reliance on Bible/Qur’an quotations

The manuscript includes lengthy scripture citations, which risk shifting the paper from academic to theological writing.

Issues

PLOS ONE requires scientific—not doctrinal—analysis.

Quotations crowd the results section and reduce analytic clarity.

Religious texts should be interpreted sociologically, not reproduced at length.

Q6: Ethical statement needs correction

The manuscript states that no minors participated, yet Table 1 includes children with cancer and caregivers of children. This is a serious inconsistency. Clearly explain assent and parental consent procedures for minors.

✔Q7. Table 1 errors

The table header does not specify whether the N refers to: patients only OR caregivers only. total participants (patients + caregivers),

Why this matters

Percentages cannot be interpreted if the denominator is unclear. Table 1 needs amendment

“N = X cancer patients”

“N = Y caregivers”

And ensure percentages are calculated from the correct denominator.

Q8: How did the authors ensure that data saturation was truly achieved, given the limited number of patient interviews and the heterogeneity of religious, gender, and contextual backgrounds?”

Reviewer #2: I think the inclusion of religious leaders muddies the paper and takes away from the focus. I would suggest getting rid of the religious leaders and discussions about religious texts and focus on what the participants say about religion and religious texts. I think that is ultimately what the author wants to do in this paper. I don’t think the author HAS to cite my work, but I do specifically write on women with metastatic breast cancer and religion. My work may be more similar to the author’s goals more so than most of the people s/he cites.

6. PLOS authors have the option to publish the peer review history of their article (what does this mean?). If published, this will include your full peer review and any attached files.

Reviewer #1: **Yes:** Fekadu Abera Kebede

Reviewer #2: No

---

## [Author Response · Author response to Decision Letter 1]

26 Mar 2026

Response to Reviewers

We sincerely thank the Editor and the Reviewers for their careful reading of our manuscript and for their insightful and constructive comments. We appreciate the opportunity to revise and improve our work. We have carefully considered all suggestions and have revised the manuscript accordingly. Below, we provide a detailed, point-by-point response to each comment. Although we have revised or reworked the entire sections of the manuscript, all the major changes are highlighted in Yellow.

Reviewer #1: The manuscript addresses an underexplored dimension of cancer care in Ethiopia—religious interpretations and meaning-making among patients and caregivers. The topic is important and culturally relevant. However, the paper requires major revisions before it can meet the scientific and methodological. Key concerns relate to conceptual clarity, methodological rigor, analytic depth, and internal consistency.

Q1: Conceptual Framework: Not Well Defined

Although the study uses phenomenology, there is no clear theoretical or analytic framework guiding how “religious meaning-making” is conceptualized.

Q2. Why this is a problem: The manuscript repeatedly uses terms such as punishment, test of faith, sovereign plan, supernatural, etc., but does not explain how these constructs are defined or distinguished in existing scholarship.

Q3.The analysis risks becoming descriptive rather than theoretically grounded.

Response: We thank the reviewer for this important observation. We agree that the initial version of the manuscript did not sufficiently articulate the conceptual and theoretical foundations guiding our analysis of religious meaning-making, especially in the context of illness causation and suffering.

In response, we have substantially revised the Introduction and Discussion sections to clarify the analytic framework underpinning the study. Specifically, we now explicitly situate our analysis within:

1. Kleinman’s explanatory model framework, to conceptualize religious interpretations as culturally shaped accounts of illness causation and suffering;

2. Social constructionist perspectives, to explain how shared religious meanings are produced and sustained within communities;

3. Medical anthropological concepts of personalistic and naturalistic etiologies, to distinguish spiritual and biomedical explanations of illness; and

4. Spiritual coping frameworks, to interpret how beliefs such as “test of faith” or “divine plan” function in emotional regulation and resilience.

We have also added a dedicated subsection in the Methods/Conceptual Framework section explaining how key constructs (e.g., punishment, test of faith, divine plan, supernatural causation) are defined and analytically distinguished based on existing scholarship. These conceptual distinctions now guide our data coding and interpretation process.

Furthermore, the Discussion section has been revised to explicitly link empirical findings to these theoretical perspectives, strengthening the analytical depth and reducing purely descriptive interpretation as rightly indicated by the reviewer.

Required Revision

Reviewer #1 comment: Provide a conceptual model linking phenomenology, meaning-making, and religious explanatory models.

Clarify whether the study draws from: Kleinman’s explanatory models, Berger & Luckmann’s social construction, Personalistic/naturalistic etiologies, Spiritual coping frameworks

Response 1: We thank the reviewer for this valuable comment. We have revised the Introduction to clarify the theoretical foundations of the study and to present an integrated conceptual framework linking phenomenology, meaning-making, and religious explanatory models. Specifically, we now situate the study within Kleinman’s explanatory model framework, Berger and Luckmann’s social constructionist perspective, Foster’s personalistic etiologies, and spiritual coping theory (pp. 7, Para. 1). This addition clarifies how these perspectives guide our analysis.

Reviewer #1 comment: Q4. Methodological transparency is insufficient.

Several methodological elements require clarification or strengthening.

1. Sampling strategy is not sufficiently justified.

Why were only 6 cancer patients included but 33 caregivers?

Phenomenology typically emphasizes deep exploration with fewer participants, not imbalanced large samples.

2. No explanation of saturation.

How was thematic saturation reached?

3. Translation procedure insufficiently described.

Back-translation?

Were multiple coders involved?

4. Data analysis lacks rigor.

No clearly mention of software, coding reliability, reflexivity, or audit trails.

Response: We thank the reviewer for highlighting the need for greater methodological transparency.

1. We have substantially revised the Methods section to clarify the sampling strategy and justify the sample composition (pp. 10, Para. 1).

“Although the initial aim was to recruit a comparable number of patients and caregivers, the study included fewer patients due to illness severity, as most presented at advanced stages, treatment schedules, and fatigue during treatment administration, particularly during and after chemotherapy. In contrast, caregivers were more accessible, interested, and actively involved in care decisions and spiritual practices. Given the study's focus on relational meaning-making within families, caregivers were retained as key participants.”

2. We have also added a detailed description of how thematic saturation was assessed (pp. 12, Para, 2),

“Participants’ recruitments continued until data saturation was reached, a point at which additional interviews yielded limited new insights and the dataset supported a coherent interpretive account [39]. Data saturation was assessed through ongoing review of interview transcripts and emerging codes. Saturation was reached after approximately 41 interviews with patients, caregivers, and religious figures, at which point further data collection was discontinued.”

3. Strengthened the translation and verification procedures (p. 11-13), and expanded the data analysis section to include information on coding procedures, use of software, reflexivity, and audit trails (pp. 12-13). These revisions enhance the rigor and transparency of the study.

Q 5: Reviewer Comment:

Over-reliance on Bible/Qur’an quotations. The manuscript includes lengthy scripture citations, which risk shifting the paper from academic to theological writing. PLOS ONE requires scientific—not doctrinal—analysis. Quotations crowd the Results section and reduce analytic clarity. Religious texts should be interpreted sociologically, not reproduced at length.

Response: We thank the reviewer for this important and constructive comment. We agree that the initial version of the manuscript relied too heavily on lengthy scriptural quotations, which may have shifted the focus from sociological analysis to theological exposition.

In response, we have carefully revised the Results and Discussion sections to substantially reduce the length and frequency of scriptural citations, focusing only on how participants used and interpreted the texts in meaning-making processes of their situations, rather than to present scriptural tests. Extended quotations have been removed or condensed and replaced with concise paraphrases that highlight their sociological significance rather than their doctrinal content.

We have also strengthened our interpretive commentary by explicitly linking participants’ use of scripture to relevant theoretical frameworks, including explanatory models, social constructionist perspectives, and spiritual coping theory.

Q6: Reviewer Comment: Ethical statement needs correction. The manuscript states that no minors participated, yet Table 1 includes children with cancer and caregivers of children. This is a serious inconsistency. Clearly explain assent and parental consent procedures for minors.

Response: We thank the reviewer for identifying this important inconsistency. We clarify that no minors were directly recruited or interviewed in this study. All interview participants were adults (≥18 years), including adult patients, adult family caregivers, and religious leaders.

The reference to “childhood cancer” in Table 1 reflects the age category of the patient being cared for, as reported by adult caregivers, rather than the age of interviewed participants. In other words, caregivers of pediatric patients were interviewed, but children themselves were not enrolled in the study.

We have revised the Ethical Considerations and Methods sections to clearly state that only adults participated in interviews. We have also revised Table 1 and its caption to clarify that age categories refer to the patients’ age at diagnosis, not the age of respondents.

Since no minors were enrolled, assent procedures were not required. Written informed consent was obtained from all adult participants.

Q7. Reviewer Comment:

Table 1 errors. The table header does not specify whether N refers to patients only, caregivers only, or total participants. Percentages cannot be interpreted if the denominator is unclear. Table 1 needs amendment.

Response: We thank the reviewer for this important observation. We agree that the original version of Table 1 did not clearly specify the reference population for N and percentages, which may have limited interpretability.

In response, we have revised Table 1 to explicitly indicate the number of participants in each category (patients, caregivers, and religious leaders) and to clarify the denominators used for percentage calculations. The table header and footnotes now specify whether percentages are calculated based on the total sample or relevant subgroups.

Specifically, we have included:

• The total number of participants (N = 41),

• The number of patients (n = 6), caregivers (n = 33), and religious leaders (n = 2), and

• Separate percentage calculations for each subgroup where applicable,

Q8: Reviewer Comment:

How did the authors ensure that data saturation was truly achieved, given the limited number of patient interviews and the heterogeneity of religious, gender, and contextual backgrounds?”

Response: We thank the reviewer for raising this important point regarding saturation. As has been stated earlier, we have revised the Methods section to clarify how saturation was systematically assessed. Data collection and analysis were conducted concurrently, and emerging themes were continuously compared across interviews and participant groups. Although the number of patient interviews was limited, in-depth idiographic analysis revealed strong convergence in core meaning-making patterns. Recruitment continued until successive interviews, including those from diverse religious and gender backgrounds, yielded no substantively new insights. I also assessed saturation across patients, caregivers, and religious leaders, which indicated thematic stability. These procedures are now described in detail on pp. 12.

Reviewer # 2: Comment: The inclusion of religious leaders muddies the paper and takes away from the focus. I suggest removing religious leaders and discussions about religious texts and focusing on participants’ perspectives.

Response:

We thank the reviewer for this thoughtful and constructive suggestion. We agree that the primary focus of this study is on the lived experiences and meaning-making processes of patients and family caregivers.

However, we included religious leaders as key informants to provide contextual insight into the religious discourses and interpretive frameworks that shape participants’ understandings of illness. Their perspectives helped to situate patients’ and caregivers’ narratives within broader community and institutional religious contexts.

In response to the reviewer’s concern, we have revised the manuscript to strengthen the centrality of patients’ and caregivers’ voices. Specifically, we have reduced the prominence of religious leaders’ accounts, limited their quotations, and ensured that they are used only to contextualize—rather than dominate—the main findings. We have also removed extended discussions of religious texts and shifted the analytical emphasis to how participants interpret and mobilize these texts in their own narratives.

These revisions sharpen the analytical focus of the manuscript while retaining the limited inclusion of religious leaders as contextual informants, thereby enhancing the depth and interpretive rigor of the study.

Comment: Page five first paragraph: “emphasizing on” Do you mean emphasis on?

Response: Thank you for noting this. We have corrected “emphasizing on” to “emphasis on” in the revised manuscript.

Reviewer’s Comment: I may have missed something, but I think the last paragraph before the methods section is the first time qualitatively data with religious figures is mentioned. Most (if not all) earlier discussions focus on patients and caregivers.

Response: We thank the reviewer for this important observation. We would like to clarify that religious figures were not the primary focus of this study. The central analytical emphasis remains on patients’ and caregivers’ experiences and meaning-making processes. Religious figures were included to get a perspective on how religious teachings, practices, and traditions shape and inform patients’ and caregivers’ interpretations of illness and care-seeking. In the revised manuscript, we have clarified this rationale in the Introduction and Methods sections to better reflect their supporting role in the analysis. Besides, we have reduced the prominence of religious leaders’ accounts, limited their quotations, and ensured that they are used only to contextualize rather than dominate the main findings. We have also removed extended discussions of religious texts and shifted the analytical emphasis to how participants interpret and mobilize these texts in their own narratives.

Reviewer’s Comment: On the bottom of page 5, there might be a word or two missing. As it reads now, the sentence is a fragment. The same thing with the first sentence of the first full paragraph on page 6.

Responses: Thank you for bringing this to our attention. We have carefully reviewed the indicated sections and revised both sentences to correct the fragments and ensure grammatical completeness and clarity.

Reviewer’s comment: The last sentence of the first full paragraph on page 6 only mentions caregivers. The next section talks about the participants, so I’m not sure you need this last sentence where it’s at. If you keep it, add all of the types of people interviewed.

Response: Thank you for this helpful suggestion. We agree that the placement and wording of this sentence could be misleading. In response, we have revised this sentence to include all participants interviewed in the study and clarified its connection to the following section. Alternatively, where appropriate, we have removed the sentence to improve coherence and flow.

Reviewer’s comment: I’m not sure perspectives from religious leaders fits within your study. You are focusing on people’s everyday lived experiences, not leaders’ perspectives on something.

Response: We thank the reviewer for raising this important point. We agree that the primary focus of this study is on patients’ and caregivers’ everyday lived experiences. The perspectives of religious leaders were not included as a separate analytical focus, but rather to provide contextual insight into the religious teachings, practices, and traditions that shape participants’ interpretations of illness and care-seeking.

Their inclusion was intended to enhance understanding of the broader religious and cultural environment within which patients and caregivers construct meaning, rather than to shift the analytical emphasis away from lived experience. In the revised manuscript, we have clarified this rationale and ensured that the Results section continues to prioritize patients’ and caregivers’ narratives.

Reviewer’s comment: At the bottom of page 6: Where did the nurse practitioner do this recruiting? Talking individually to her patients? At a class or seminar? A signup sheet?

Response: Thank you for requesting clarification on the recruitment process. Initial contact with potential participants was facilitated in

---

## [Decision Letter · Decision Letter 1]

22 Apr 2026

PONE-D-25-58785R1Negotiating Illness through Faith: Religious Narratives in Cancer Etiology among Patients and Caregivers in Addis Ababa, EthiopiaPLOS One

Dear Dr. Fikre,

Thank you for submitting your manuscript to PLOS ONE. After careful consideration, we feel that it has merit but does not fully meet PLOS ONE’s publication criteria as it currently stands. Therefore, we invite you to submit a revised version of the manuscript that addresses the points raised during the review process

**Please, address the issues pointed by the reviewer, they need to be clarified before the manuscript can progress to the next stage.**

We look forward to receiving your revised manuscript.

Kind regards,

Rafael Galvão de Almeida, PhD.

Academic Editor

PLOS One

**Journal Requirements:**

Reviewers' comments:

Reviewer's Responses to Questions

**Comments to the Author**

1. If the authors have adequately addressed your comments raised in a previous round of review and you feel that this manuscript is now acceptable for publication, you may indicate that here to bypass the “Comments to the Author” section, enter your conflict of interest statement in the “Confidential to Editor” section, and submit your "Accept" recommendation.

Reviewer #2: (No Response)

2. Is the manuscript technically sound, and do the data support the conclusions?

Reviewer #2: Yes

3. Has the statistical analysis been performed appropriately and rigorously? 

Reviewer #2: N/A

4. Have the authors made all data underlying the findings in their manuscript fully available?

Reviewer #2: Yes

5. Is the manuscript presented in an intelligible fashion and written in standard English?

Reviewer #2: Yes

6. Review Comments to the Author

**Reviewer #2:** Data collection: the fact that you did not have pre-existing questions regarding religion is a big deal. It is telling that your interviewees brought religion up without prompt. You may want to consider emphasizing this point more.

On page 15: I’m not sure about your discussion of transferability. What is written is very vague and doesn’t get at how/whether your study would be applicable in somewhat similar and different contexts. You may want to consider a transferability discussion in the suggestions for future research.

Page 17: I don’t think you need the box. The “theme” isn’t really a finding. It’s more of a Question. I do think a summary paragraph at the beginning of a findings section is helpful, so if you remove the box, you can mention the subthemes in a paragraph.

On page 17: it doesn’t make sense that the quote you provide was provided by multiple participants. It’s quite ok to talk about the general pattern and provide a quote that illustrates the pattern.

Page 18: who is “her”?

Page 19 second paragraph: what is “it”?

Page 19 last paragraph: What you have here is an EXCELLENT justification for why you include some commentary from religious leaders. It may help to emphasize this connection early on in the paper.

Page 20: pastor pastor’s

Page 25: I’m not understanding the “origin of cancer and its diagnosis”

7. PLOS authors have the option to publish the peer review history of their article (what does this mean?). If published, this will include your full peer review and any attached files.

Reviewer #2: No

---

## [Author Response · Author response to Decision Letter 2]

27 Apr 2026

Response to Reviewers

We sincerely thank the Editor and the Reviewers for their careful reading of our manuscript and for their insightful and constructive comments. We appreciate the opportunity to revise and improve our work. We have carefully considered all suggestions and have revised the manuscript accordingly. Below, we provide a detailed, point-by-point response to each comment. Although we have revised or reworked the entire sections of the manuscript, all the major changes are highlighted in Yellow.

Comments from Reviewer #2: Data collection: the fact that you did not have pre-existing questions regarding religion is a big deal. It is telling that your interviewees brought religion up without prompt. You may want to consider emphasizing this point more.

Response:

Thank you for this insightful observation. We agree that the spontaneous emergence of religious narratives during interviews is an important finding in itself. In response, we have revised the manuscript to clarify that religion was not included as a pre-defined focus in the interview guide. Instead, participants independently raised religious explanations and interpretations when discussing their illness experiences. We now emphasize this point more explicitly in the data collection section, as it highlights the salience and centrality of religion in participants’ meaning-making processes. See it from the revised manuscript pp. 12, highlighted in yellow.

Reviewer Comment 1

On page 15: I’m not sure about your discussion of transferability. What is written is very vague and doesn’t get at how/whether your study would be applicable in somewhat similar and different contexts. You may want to consider a transferability discussion in the suggestions for future research.

Response 1

Thank you for this insightful comment. We have substantially revised the manuscript to strengthen the discussion of transferability. Specifically, we have clarified how the findings are contextually grounded and articulated the conditions under which they may be relevant to other settings, particularly those with similar sociocultural and healthcare characteristics.

In addition, we have explicitly addressed transferability in two key sections. First, we have incorporated a more focused discussion in the Limitations section, emphasizing the context-specific nature of the findings and the potential constraints in applying them to more secular or structurally different healthcare environments. Second, we have expanded the Implications/Future Research section to outline how comparative and cross-contextual studies could further assess and extend the transferability of these findings.

We have also refined the trustworthiness section by providing a more concrete account of how transferability was supported through rich, contextualized descriptions of the study setting and participants, enabling readers to assess relevance to other contexts.

We believe these revisions address the reviewer’s concern and improve the clarity and rigor of the manuscript. See the changes from pp. 15 section: Trustworthiness of the study; pp. 37, section Implication of the study; pp.38, section Limitations of the study, all highlighted in Yellow.

Reviewer Comment 2

Page 17: I don’t think you need the box. The “theme” isn’t really a finding. It’s more of a Question. I do think a summary paragraph at the beginning of a findings section is helpful, so if you remove the box, you can mention the subthemes in a paragraph.

Response 2

Thank you for this helpful suggestion. We agree that the table did not effectively present the analytical contribution of this section. In response, we have removed the table and revised the Results section to begin with a concise summary paragraph that synthesizes the key findings.

Specifically, we now introduce the subthemes as analytically derived interpretations—namely, cancer as divine punishment, a test of faith, a manifestation of divine or sovereign will, and as an attack by evil spirits or Satan—rather than presenting them as items within a table. This restructuring allows for a more coherent and interpretive presentation of the findings and improves the overall narrative flow of the section.

We believe this revision strengthens the clarity and analytical depth of the manuscript and better aligns with qualitative reporting standards. See the changes from the results section first paragraph, pp.15, highlighted in yellow.

Reviewer Comment 3

On page 17: it doesn’t make sense that the quote you provide was provided by multiple participants. It’s quite ok to talk about the general pattern and provide a quote that illustrates the pattern.

Response 3:

Thank you for this important comment. We agree that the previous version did not clearly distinguish between the general analytic pattern and the illustrative role of individual quotations. In response, we have revised the section to first present the shared pattern identified across participants in our analytic voice, followed by a single participant quotation used strictly as an illustration of this pattern.

We have also removed phrasing that implied the quotation represented multiple participants and revised the surrounding text to improve clarity and consistency in the presentation of findings. These changes enhance the transparency and rigor of the qualitative analysis and align the manuscript with established conventions for reporting qualitative data. See the changes from pp. 17, highlighted in a yellow.

Reviewer Comment 4:

Page 18: who is “her” in this sentence? The referent is unclear.

Response 4:

Thank you for pointing out this ambiguity. We agree that the original wording was unclear regarding the referent of “her.” We have revised the paragraph to clearly distinguish between the caregiver (the participant) and the patient (her mother), ensuring consistent and precise attribution throughout the narrative. These revisions improve clarity and avoid any potential confusion in interpreting the quotation and its accompanying analysis. See the changes from pp. 19, highlighted in a yellow.

Reviewer Comment 5:

Page 19 second paragraph: what is “it”?

Response:

Thank you for this comment. We agree that the previous use of the pronoun “it” was unclear. We have revised the sentence for clarity by replacing the ambiguous reference with a precise subject. The revised version now reads: “These interpretations linked illness and suffering to perceived moral transgressions, reinforcing understandings of cancer as spiritually consequential.” This revision improves grammatical clarity and strengthens the conceptual precision of the paragraph.

Reviewer Comment 6:

Page 19 last paragraph: What you have here is an EXCELLENT justification for why you include some commentary from religious leaders. It may help to emphasize this connection early on in the paper.

Response 6:

Thank you for this valuable suggestion. We agree that the role of religious leaders and broader religious institutions in shaping illness meanings needed to be made more explicit at an earlier stage. In response, we have revised the opening of the “Religious Narratives of Illness” section to clearly indicate that participants’ interpretations of cancer are not only individually constructed but also shaped and reinforced through broader social and institutional religious influences.

Specifically, we have added a framing paragraph that highlights the role of religious leaders, communal worship, and pastoral counseling in reinforcing moral and spiritual interpretations of illness. This revision strengthens the conceptual coherence of the section and clarifies the importance of social and institutional contexts in shaping participants’ meaning-making processes.

Reviewer Comment 7

Page 20: pastor pastor’s

Response 7

Thank you for this comment and we have corrected the phrasing for grammatical accuracy and clarity by removing the repetition and ensuring correct possessive usage. The sentence has been revised to clearly indicate that the explanation is attributed to a single pastor, thereby improving readability and precision in the presentation of the findings.

Reviewer Comment 8

Page 25: I’m not understanding the “origin of cancer and its diagnosis”

Response: 8

Thank you for this comment. We agree that the previous wording was unclear. We have revised the sentence for clarity and conceptual precision. The phrase “origin of cancer and its diagnosis” has been removed and replaced with “cancer” to better align with the focus on illness narratives and meaning-making. The revised sentence now reads: “Another prominent illness narrative interpreted cancer as part of a divine sovereign plan.” We believe this improves clarity and strengthens the analytical framing of the findings.

---

## [Editor Report · Decision Letter 2]

30 Apr 2026

Negotiating Illness through Faith: Religious Narratives in Cancer Etiology among Patients and Caregivers in Addis Ababa, Ethiopia

PONE-D-25-58785R2

Dear Dr. Fikre,

We’re pleased to inform you that your manuscript has been judged scientifically suitable for publication and will be formally accepted for publication once it meets all outstanding technical requirements.

Kind regards,

Rafael Galvão de Almeida, PhD.

Academic Editor

PLOS One

Additional Editor Comments (optional):

In the added line "Questions related..." I'm just curious if the original idea was to study religious influence on the patients or if the study's objective was changed after this was observed, you can refer to it in first person. Also, please include an "Acknowledgements" section for any help you had with the study and the two reviewers.
---

## [Editor Report · Acceptance letter]

PONE-D-25-58785R2

PLOS One

Dear Dr. Fikre,

I'm pleased to inform you that your manuscript has been deemed suitable for publication in PLOS One. Congratulations! Your manuscript is now being handed over to our production team.

Kind regards,

on behalf of

Dr. Rafael Galvão de Almeida

Academic Editor

PLOS One